

# The life cycle of nocturnal low-level clouds over southern West Africa analysed using high-resolution simulations

Bianca Adler[1], Norbert Kalthoff[1], and Leonhard Gantner[1]

[1]Institute of Meteorology and Climate Research, Karlsruhe Institute of Technology (KIT), Germany

*Correspondence to:* Bianca Adler (bianca.adler@kit.edu)

**Abstract.** We performed a high-resolution numerical simulation to study the life cycle of extensive low-level clouds which frequently form over southern West Africa during the monsoon season. This study was made in preparation for a field campaign in 2016 within the Dynamics-aerosol-chemistry-cloud interactions in West Africa (DACCIWA) project and focuses on an area around the city of Savè in southern Benin. Nocturnal low-level clouds evolve a few hundred metres above the ground around

the same level as a distinct low-level jet. Several processes are found to determine the spatio-temporal evolution of these clouds including (i) significant cooling of the nocturnal atmosphere due to horizontal advection with the south-westerly monsoon flow during the first half of the night, (ii) vertical cold air advection due to gravity waves leading to clouds in the wave crests and (iii) enhanced convergence and upward motion upstream of existing clouds that trigger new clouds. The latter is caused by an upward shift of the low-level jet in cloudy areas leading to horizontal convergence in the lower part and to horizontal divergence

in the upper part of the cloud layer. Although this single case study hardly allows for a generalisation of the processes found, the results added to the optimisation of the measurements strategy for the field campaign and the observations will be used to the test the hypotheses for cloud formation resulting from this study.

## 1   Introduction

During the West African monsoon season, nocturnal low-level stratiform clouds frequently form over southern West Africa

with a cloud base of only few hundred metres above ground (Knippertz et al., 2011; Schrage and Fink, 2012; Schuster et al., 2013; van der Linden et al., 2015). From synoptic observations and various satellite products van der Linden et al. (2015) derived a climatology of low-level clouds over southern West Africa for the monsoon seasons from 2006 to 2011. The affected area covers approximately 800,000 km$^2$. Low-level clouds frequently form shortly after sunset along the coast and upstream of the Mampong Range in Ghana and the Oshogbo Hills in Nigeria, spread during the night and dissipate in the late morning

or early afternoon, forming convective clouds (Schuster et al., 2013; van der Linden et al., 2015).

The mechanisms controlling the formation and maintenance of the low-level clouds were investigated by Schrage and Fink (2012) using observations at Nangatchori (Benin, 9.70° N, 1.68° E, 434 m above mean sea level (m MSL)) and by Schuster et al. (2013) who simulated the 2006 monsoon season and analysed atmospheric conditions averaged for clear and cloudy nights. Based on these few studies, current theories relating to low-level cloud formation suggest that these clouds evolve

in connection with a nocturnal south-westerly low-level jet (LLJ), which forms regularly in this region and is linked to the





Saharan heat low (e.g. Parker et al., 2005; Lothon et al., 2008; Abdou et al., 2010). The jet axis is several hundred metres above ground, i.e. generally around the same level as the low-level clouds. In the simulations of Schuster et al. (2013), the LLJ is on average several m s$^{-1}$ stronger during cloudy nights than during clear nights. They propose that the relevance of different processes to low-level cloud formation varies depending on the region. While shear-generated turbulent vertical mixing of

moisture underneath the LLJ is a main process close to the coast, orographically induced lifting upstream of higher terrain is more important farther inland. In both regions, horizontal cold air advection is strong in and underneath the LLJ layer and radiative cooling at the cloud top helps maintain the low-level clouds once they have formed. Schrage and Fink (2012) propose that vertical mixing of moisture due to shear-generated turbulence underneath the LLJ is the major process for cloud formation far inland at Nangatchori. However, no answers were given to the questions why many nights with an LLJ remain clear and

why the LLJ does not form during some nights with low-level clouds.

The existing studies suggest that a lot of effort is still needed to identify controls of the life cycle of the low-level clouds and to improve the understanding of the involved physical processes. In particular, there is a large demand for high-quality observations in this region for process studies and model validation. To meet this demand, a field campaign was conducted in the 2016 monsoon season in southern West Africa within the framework of the Dynamics-aerosol-chemistry-cloud interactions

in West Africa (DACCIWA) project (Knippertz et al., 2015a). In the regions favourable for low-level clouds (van der Linden et al., 2015), three supersites were installed at Kumasi (Ghana), Savè (Benin) and Ile-Ife (Nigeria).

In preparation for this field campaign, we performed high-resolution simulations with the COSMO model for the area around the supersite at Savè (8.00° N, 2.43° E, 166 m MSL), i.e. about 200 km inland from the Guinea Coast (Fig. 1). At this supersite, various remote-sensing and in-situ systems, including the KITcube mobile observation platform (Kalthoff et al., 2013), were

operated during the field campaign. This study aims at identifying possible controls of the life cycle of the nocturnal low-level clouds in the simulation. This results could then be used to optimize the measurement strategy for the field campaign. Contrary to Schuster et al. (2013) who simulated a large part of southern West Africa for the whole monsoon season of 2006, we performed a case study for a smaller domain around the supersite at Savè with a significantly higher horizontal and vertical resolution. This set-up was chosen to resolve the vertical structure of the nocturnal boundary layer and the LLJ as well as

small-scale processes possibly relevant to the low-level clouds.

The paper is structured as follows: the next section covers the model set-up, followed by a description of the simulated life cycle of low-level clouds in Section 3. In Section 4, we analyse the atmospheric conditions and processes relevant to low-level clouds for different phases of the night. Section 5 presents a discussion and Section 6 provides a summary and conclusions.

## 2 Model set-up

For this study, we use the Consortium for Small Scale Modeling (COSMO) model, version 5.1, which is a fully compressible non-hydrostatic regional weather forecast model (Schättler et al., 2014) used for operational weather forecasting as well as for scientific applications down to high resolutions. The model run was performed with a horizontal grid spacing of around 500 m (0.005°) and a hybrid system with 80 layers up to 22 km with 33 layers below 1.8 km. For comparison, Schuster et al. (2013)





used a horizontal grid spacing of 3 km and 70 vertical layers, of which 20 were below 1.8 km. The horizontal differencing in the COSMO model is done on a latitude/longitude grid using an Arakawa C-grid and a generalised terrain-following height coordinate is implemented in vertical direction. For turbulent diffusion, we use a 3-D turbulence parametrisation based on the extended Smagorinsky model according to Herzog et al. (2002) and also applied by Fiori et al. (2010) and Gantner et al. (2016).

In this parametrisation turbulence diffusion coefficients depend on horizontal and vertical grid size, stability and turbulent kinetic energy, which is retrieved from a prognostic equation. Due to the high horizontal resolution, the model is convection-permitting and therefore convection parametrisation is turned off in this simulation. At lower horizontal resolutions, sub-grid-scale clouds are considered in the radiation scheme after Ritter and Geleyn (1992) by applying either a relative humidity or a statistical criterion. However, these criteria likely are not applicable at such high horizontal resolution as chosen in this

study. Sensitivity tests show that the consideration of sub-grid-scale clouds in the radiation scheme delays the transition from stratiform clouds to convective clouds by several hours, while the impact on the qualitative characteristics of the nocturnal low-level clouds is small. In this simulation, only grid-scale clouds are considered in the radiation scheme and the scheme is called every 15 min. Using an online trajectory module implemented in COSMO (Miltenberger et al., 2013), trajectories are started hourly at 7.5° N between 1.5° and 2.5° E at various levels below 2000 m MSL.

For initialisation and boundary conditions of the atmospheric part of the model, a convection-permitting simulation with 2.8 km horizontal grid spacing performed with the COSMO model for the northern half of Africa for the whole year of 2006 (Maurer et al., 2016) is used. An advantage of using this simulation is that the lower boundary, particularly the soil moisture, is tuned and long spin-up times are unnecessary. As the 2.8-km simulation is initialised on 1 January 2006 and the model domain is very large (about 2500 x 1800 grid points), the simulated atmospheric conditions on a specific day do not necessarily agree

with the observations. However, the overall characteristics of the simulated data are realistic and precipitation statistics match well with Tropical Rainfall Measuring Mission (TRMM) data (Maurer et al., 2016).

   We examine the 2.8-km simulation for periods during the monsoon season when the conditions in the area of interest are characterised by conditions assumed to be favourable for the evolution of low-level clouds (Schrage and Fink, 2012). This means we are looking for periods with a south-westerly monsoon flow, a strong nocturnal low-level jet and high relative

humidity at low levels, which are free of disturbances by large-scale events such as mesoscale convective systems and free of precipitation. Of these periods, we choose the night from 3 to 4 August for analysis. The run is initialised at 1200 UTC on 3 August and the boundary conditions are updated every hour. For the analysis, we use 15-min model output on $z$-levels. Times are given in UTC, the local standard time in Benin being UTC plus 1 h. The model domain ranges from 6.5° to 9° N and 1° to 3.5° E and contains 501 x 511 grid points (Fig. 1). To prevent boundary effects, the analysis is confined to a smaller area

with about 100 km distance to the upstream (southern and western) boundaries. In the area used for the analysis, the terrain gradually increases from south to north, with Savè being about 100 km south-west, i.e. upstream, of the Oshogbo Hills, which rise up to 500 m MSL. To assess the impact of different terrain heights on the life cycle of low-level clouds, we investigate two areas of 40 km x 40 km in detail: the SAVE area, which is centred on the location of the supersite at Savè, is characterised by a mean terrain height of 156 m MSL and the HILLS area is directly upstream of the peaks of the Oshogbo Hills with a mean

terrain height of 322 m MSL.



## 3  Life cycle of low-level clouds

To obtain information about the spatial distribution and temporal evolution of low-level clouds, we accumulate the liquid water content below 1000 m MSL (Figs. 2 a, b) and average the liquid water content for both areas SAVE (Fig. 2c) and HILLS (Fig. 2d). Throughout the night, the spatial extent as well as the intensity of low-level clouds varies, allowing to distinguish different phases.

Before about 2200 UTC, the analysed area is cloud-free (Phase 0). First low-level clouds form after around 2200 UTC in HILLS and after midnight in SAVE (Phase 1, Figs. 2c, d). During this phase, the low-level clouds are thicker and more extended over the higher terrain in the north-eastern part of the analysed area (Fig. 2a). In the other parts of the area, clouds remain rather patchy and thin. After 0300 UTC, thick low-level clouds spread towards the south-west, thus increasing the liquid water content in SAVE (Phase 2, Figs. 2b, c). After around 0500 UTC, the thick clouds withdraw somewhat to the north-east, causing the averaged liquid water content in SAVE to decrease (Fig. 2c). During this phase, the clouds in HILLS are more intense than during Phase 1 (Fig. 2d). Considering the different terrain heights, cloud-base heights above sea level are very similar in both areas (Figs. 2c, d). With the onset of surface heating, the cloud base rises after 0800 UTC and convective clouds form.

## 4  Atmospheric conditions and processes relevant to low-level clouds

The differences in the temporal and spatial evolution of clouds in SAVE and HILLS suggest that different processes are relevant in the two areas. To understand the evolution of low-level clouds, we analyse in detail the atmospheric conditions and the processes affecting these conditions during the different phases. For this purpose, we calculate tendencies of potential temperature and moisture from the 15-min model output and average them for the SAVE (Fig. 3) and HILLS areas (not shown). When using the term moisture, we are referring to specific humidity. To derive the different budget terms, we determine horizontal and vertical advection for each grid point and each level and average over the area afterwards. Note that parts of the contributions by advection result from inclined isentropes and isohumes over the smoothly sloping terrain. As contributions of sub-grid-scale mixing, divergence of net radiation or phase changes to the tendencies, cannot be quantified in the simulation due to unavailable output, residuals result. The residuals are determined by simply subtracting the advection terms from the tendencies (Figs. 3d, h). Although the residuals are influenced by subtracting the instantaneous advection terms from the tendencies calculated from the 15-min output, the magnitude and sign of the residuals provide some qualitative information: for example, the residual for the heat budget is found to be very large near the surface during Phase 0 when a nocturnal inversion evolves due to surface cooling (negative sensible surface heat flux) and in the morning when the convective boundary layer forms due to heating of the surface (positive sensible surface heat flux). During Phases 1 and 2, the budgets are dominated by the contributions of the advection terms.

25





### 4.1 Before the evolution of low-level clouds (Phase 0)

The evolution of the stably stratified nocturnal boundary layer is significantly influenced by strong cooling and moistening in the first half of the night (shown for SAVE in Figs. 3a, e), leading to a rapid increase of relative humidity throughout the lower atmosphere (shown for SAVE in Fig. 4).

Most of the cooling and moistening occurs between 2030 and 2300 UTC in SAVE (Figs. 3a, e) and between 2100 and 2300 UTC in HILLS (not shown). Due to meridional averaging, cooling and moistening appears to be rather smooth. The analysis of spatial distributions of atmospheric variables (not shown), however, reveals that the temperature and humidity changes are associated with a cool and moist air mass which propagates northwards showing a front-like, sharp, roughly west-east oriented boundary on its leading edge. Behind the front, relative humidity increases to more than 90 % (Fig. 4), making the atmosphere

favourable for the formation of clouds.

    As the front seems to play an important role for the formation of the low-level clouds, we investigate the 2.8-km simulation during the months of July and August 2006 for the occurrence of this front. In particular, we inspect relative humidity at 950 hPa (around 500 m MSL). The front occurs regularly when an undisturbed south-westerly monsoon flow prevails along the coast and further inland. Figure 5 shows the relative humidity at 950 hPa averaged for the two months. The front evolves

along the coast after noon, which is reflected by a strong gradient in relative humidity between the relatively cool maritime air mass over the Gulf of Guinea and warmer air over land (Fig. 5a). In general, moisture increases from south to north, which is related to strong evaporation over land. Within this large-scale moisture difference a local maximum exists along the front. This is caused by moisture convergence and upward transport of moisture from close to the surface when the monsoon flow decelerates due to surface friction when reaching the coast. During the afternoon, the front is rather stationary located about 30

km inland along the coast of eastern Ghana, Togo and western Benin. Further to the east, the front is more diffuse and locates farther away from the coast. After 1600 UTC, the front starts penetrating inland and reaches SAVE at around 2100 UTC on the average (Fig. 5b), which agrees well with the conditions of the case study (Figs. 3a, e and 4). During the subsequent hours, the front becomes more diffuse on the average, but continues to propagate inland.

### 4.2 First clouds (Phase 1)

During the passage of the front, thin clouds already form in HILLS (Fig. 2d). Trajectories reveal that air parcels rise by up to 200 m when following the smoothly rising terrain to the north-east, leading to continuous cooling, an increase of relative humidity and the formation of clouds (Figs. 6a, b). This is in agreement with heat-budget calculations, which reveal that horizontal and vertical cold air advection is responsible for the maintenance of the clouds in HILLS (not shown). This suggests that orographically induced lifting is most important for cloud formation in the north-eastern part of the analysed area.

Once the front has passed SAVE, it takes another hour of additional cooling and moistening, mainly between about 550 and 800 m MSL and primarily due to vertical advection, before first clouds evolve (Fig. 3). Below this layer, cooling due to horizontal advection persists and weak horizontal dry air advection occurs, which agrees with the general large-scale south-north increase of moisture. Horizontal advection after the passage of the front is largely related to the LLJ, which evolves after





2200 UTC in SAVE and reaches up to 12 m s$^{-1}$at around 700 m MSL (contours in Fig. 2c). Vertical advection is caused by mean upward motion in SAVE, which transports moist and cool air from below. Besides orographically induced lifting, which is less strong in SAVE than in HILLS due to different terrain gradients, gravity waves contribute to the upward motion in the nocturnal atmosphere (Fig. 6). Lifting and cooling cause the highest relative humidity in the wave crests. During Phase 1, saturation is reached in wave crests and clouds form in broken band-like structures perpendicular to the mean flow in some regions in the south-western part of the analysed area (Figs. 2a and 6c, d). To further investigate the gravity waves, we produced cross sections through the centre of SAVE of horizontal wind, vertical wind and potential temperature aligned along the mean wind direction (Fig. 7). Gravity waves evolve between about 400 and 1000 m MSL with vertical motion being strongest in the stably stratified layer between around 600 and 800 m MSL. Fast Fourier Transformation reveals a phase lag of roughly 90° between the waves of potential temperature and vertical wind speed, which is a characteristic of gravity waves, and dominant wave lengths for both variables ranging between 5 and 10 km (not shown). Due to low static stability and/or high wind shear, the Richardson number is smaller than the critical value of 0.25 below and above the height of the LLJ maximum (not shown) and sub-grid-scale turbulence (horizontal lines in Fig. 7b) could contribute to an upward transport of moist and cool air. However, the small residuals of heat and moisture budgets during this phase (Figs. 3d, h) suggest that this process is not very effective.

## 4.3 Intensification and spatial expansion of clouds (Phase 2)

As described in Section 3, thick extended low-level clouds are confined to the north-eastern part of the analysed area, i.e. downstream of SAVE, during Phase 1. After 0300 UTC, thick clouds suddenly start to form in the south-western part as well and affect SAVE (Figs. 2b, c). We assume that two processes are mainly responsible for this: (i) during Phases 0 and 1, the atmosphere in and below the clouds is continuously cooled by vertical and horizontal advection, respectively (Figs. 3a, b, c), while changes in moisture are small (Figs. 3e, f, g). Consequently, relative humidity increases continuously in the respective layer (Fig. 4). (ii) In this environment, enhanced vertical cold air advection leads to the evolution of thick clouds in SAVE, initiating Phase 2 (Figs. 3a, c). The enhanced vertical cold air advection is mainly caused by an increase of the mean upward motion (Fig. 8a). This process is also reflected by the trajectories of air parcels started at 0200 UTC (Figs. 6c, d): air parcels rise by about 100 m in SAVE (between around 60 and 80 km from starting point in Fig. 6d), leading to saturation (Fig. 4) and clouds (Figs. 2b, c and 6c, d). This strong rise of trajectories in SAVE does not occur before (Figs. 6a, b).

The sudden stronger upward motion cannot be caused by an amplification of gravity waves, because their intensity remains about the same. Instead, it is related to a significant increase in horizontal convergence below about 800 m MSL, i.e. in the lower part of the clouds and below them (Fig. 8b). Horizontal convergence in SAVE is related to a modification in stratification and horizontal wind profiles in areas with thick clouds compared to cloud-free areas: static stability is significantly lower within clouds than at the same level outside the clouds, as is visible along the cross section through the centre of SAVE (Fig. 8c). At the same time, static stability at the cloud top is higher compared to the same level in cloud-free areas. This shift in stratification within the clouds is probably caused by enhanced turbulent kinetic energy (white lines in Fig. 8d), latent heat release due to condensation (positive residual of the temperature budget in the lower part of the cloud layer in Fig. 3d), radiative cooling at cloud top (negative residual of the temperature budget in the upper part of the cloud layer in Fig. 3d) and upward motion in





the stably stratified atmosphere (Fig. 8a). The height of the LLJ maximum shifts towards the layer of maximum static stability (Fig. 8d). Consequently, horizontal wind speed increases near cloud top, while it decreases within the clouds. This results in horizontal convergence upstream of the clouds in the layers in the lower part and below the clouds and horizontal divergence in the layer near cloud top (Fig. 8b). When relative humidity has sufficiently increased due to process (i), process (ii) becomes

active and triggers new thick clouds upstream of existing ones.

The thick clouds spread to the south-west and at 0500 UTC, the whole area of SAVE is covered by thick clouds (Figs. 2b, c). This also marks the maximum south-west extension of thick clouds. The relative humidity is generally lower by several % south-west of SAVE, which is likely related to the large-scale decrease of moisture from north to south. With an increasing coverage of thick clouds in SAVE - reflected by an increasing averaged liquid water content (Fig. 2c) - horizontal convergence,

upward motion and, consequently, vertical cold air advection decrease again in this area (Figs. 3c and 8a, b). After around 0500 UTC, thick clouds gradually retreat to the north-east. When parts of SAVE become cloud-free at around 0630 UTC, process (ii) becomes active again with horizontal convergence and upward motion increasing (Figs. 8a, b) and some thick clouds forming. However, vertical cold air advection is much weaker than before (Fig. 3c), as the lapse rate in the former cloud layer is reduced due to turbulent mixing, latent heat release, radiative cooling and upward motion. As a consequence, horizontal warm air

advection due to inclined isentropes is dominant and causes net warming and a reduction of relative humidity.

A convective boundary layer evolves after sunrise at 0600 UTC (Figs. 3a, d). When it reaches the bottom of the cloud layer in SAVE at around 0800 UTC, the cloud base starts to rise and eventually a transition to convective clouds occurs (Fig. 2c).

## 5   Discussion

The processes relevant to cloud formation take place on different scales: the increase of relative humidity due to horizontal cold

air advection from the coast is related to the monsoon flow and the LLJ, i.e. to large-scale processes. Clouds are triggered by orographically induced lifting, gravity waves and horizontal convergence upstream of existing clouds, which occur on a scale of a few hundred to a few tenths of kilometres. In addition, turbulent sub-grid-scale mixing is important for the upward shift of the LLJ in cloudy areas. The capability of models to resolve any of these processes, of course, depends on the model-grid spacing. This has to be kept in mind when comparing these results to other studies.

The characteristics of the LLJ as well as the temporal evolution and spatial distribution of the low-level clouds are in general agreement with previous studies: the strength and height of the LLJ are comparable to the LLJ characteristics during cloudy nights reported by Schrage and Fink (2012) and Schuster et al. (2013) considering that the LLJ and cloud base in this simulation are roughly constant in height above sea level. The low-level clouds in this simulation first form directly upstream of the Oshogbo Hills due to orographically induced lifting and then spread to the south-west, i.e. to the upstream side. The

feature is also visible in satellite images (van der Linden et al., 2015), which supports our hypothesis that enhanced upward motion upstream of existing thick clouds triggers new clouds. We find that this process is most relevant to the evolution of thick, extended low-level clouds in SAVE. According to our knowledge, this process has not been reported before. In the simulations of Schuster et al. (2013) clouds form upstream of the Oshogbo Hills due to orgraphically induced lifting and intensify during the





night. However, no clear extent to the south-west upstream side can be distinguished. This could be related to the coarser grid spacing of the model or to different parameterisations. On the one hand, the modification of the horizontal wind field due to the grid-scale clouds is related to model-grid spacing. On the other hand, the upward shift of the layer with maximum static stability and, hence, of the LLJ is at least partly caused by sub-grid-scale turbulent mixing, which depends on the parameterisation in the model. The analysis of the sensitivity of this trigger mechanism to different turbulence paramterisations and grid spacing is beyond the scope of this study, but could be addressed in future studies.

Besides the formation of low-level clouds over land upstream of orography, Schuster et al. (2013) and van der Linden et al. (2015) find that low-level clouds form early along the Guinea Coast and then spread inland. The model-domain size and location in this study do not allow to study the latter process in detail. Nevertheless, we hypothesise that conditions at the coast and upstream of orography are generally earlier favourable for cloud formation than in other areas over land. In the course of the night, conditions become more favourable in the other areas as well - likely due to horizontal cold air advection and radiative cooling - leading to an extent of low-level clouds from the coast to the north as well as to the south-west upstream of orography.

In this simulation, gravity waves in the LLJ layer generate clouds. To exclude that the gravity waves are caused by numerical effects and enter the domain through the boundaries, we performed another model run with a doubled domain size. The southern boundary for this run is located over the Gulf of Guinea. It is evident that the gravity waves are generated over land within the domain and do not enter through the domain borders and travel through the domain. The gravity waves evolve under stable stratification and significant vertical wind shear, i.e. under conditions in which gravity waves are common features (e.g. Fritts et al., 2003; Newsom and Banta, 2003; Wang et al., 2013). Besides vertical wind shear, some of the gravity waves seem to be linked to orographic features, although obstacle heights upstream of SAVE are on the order of 50 m only.

The front which exists along the coast of southern West Africa and moves inland in the evening is a regular feature in the 2.8-km simulation. As far as we know, it has not been reported before in this region, which is maybe due to a lack of observations or to lacking analysis of high-resolution simulations. On the other hand, the formation of a sea breeze in this region was documented by satellite images (e.g. Cautenet and Rosset, 1989) and investigated using observations in Cotonou (Bajamgnigni and Steyn, 2013) and at several stations along the coast of Nigeria (Abayomi et al., 2007). The south-westerly monsoon flow during the wet season often prevents a reversal to offshore flow along the coast of southern West Africa in the evening (Bajamgnigni and Steyn, 2013) and may make the sea breeze indistinguishable from the large-scale flow as known from other regions (overview by Crosman and Horel, 2010). For this reason, we use the term "front" rather than "sea-breeze front" to describe the boundary between the cool maritime and the warm continental air masses, which evolves along the coast during the day. The temporal characteristics of the front, i.e. its stationarity during the day and its inland penetration in the late afternoon and evening, are highly similar to a study by Grams et al. (2010). They investigated a sea-breeze front on the coast of Mauretania, which is stationary during the day and moves several hundred kilometres inland in the evening. In the opinion of the authors the stationarity results from the balance between horizontal advection of cool maritime air and turbulent mixing in the convective boundary layer. When turbulence decays in the evening, the sea-breeze front moves inland. Other authors





report cool sea-breeze surges in northern (Garratt and Physick, 1985) and south-eastern Australia (Taylor et al., 2005) which penetrate up to 500 km inland at night with moderate onshore flow.

## 6 Summary and Conclusions

A high-resolution COSMO simulation was performed for southern West Africa during the monsoon season in order to identify possible controls of the life cycle of nocturnal low-level clouds. The analysis was made in preparation for the DACCIWA field campaign and focuses on the area around the city of Savè (SAVE area), the location of one of the three supersites. The conditions during the case study are typical of nights with an undisturbed south-westerly monsoon flow prevailing in the area of interest.

Based on the simulation data we hypothesise that several processes are relevant to cloud formation; some of these are illustrated in Fig. 9. A sudden increase of relative humidity occurs in the first half of the night. This is related to a front which forms along the Guinea Coast during the day between the cool maritime and the warm continental air masses and moves inland in the late afternoon and evening. Subsequent cooling mainly due to horizontal advection with the LLJ and vertical advection results in the formation of clouds. Vertical cold air advection is related to orographically induced lifting (process 1 in Fig. 9) as well as to gravity waves which form in the stably stratified atmosphere around the level of the LLJ (process 2 in Fig. 9). In areas covered by clouds, the height of the LLJ maximum shifts to the top of the clouds, resulting in low-level horizontal convergence and upward motion upstream of the clouds. This triggers new clouds upstream of the existing ones, when saturation is reached (process 3 in Fig. 9). Latent heat release due to condensation and radiative cooling at the cloud top likely contribute to cloud evolution as well, but are not the main mechanisms according to the heat and moisture budgets.

The simulation reveal some interesting processes contributing to the life cycle of low-level clouds over southern West Africa, which have not been reported before. The results contributed to an optimized measurement strategy for the DACCIWA field campaign, as they emphasize the importance of measuring advection of heat and moisture, turbulence profiles, profiles of horizontal wind speed with high vertical and temporal resolution to resolve the structure and evolution of the LLJ and vertical motion in the LLJ layer to detect gravity waves. The observations will then be used to verify and test the hypotheses for cloud formation gained from this study.

Of course, it is difficult to draw any general conclusions with respect to the representativeness of the processes from this case study. Nevertheless, we think that the processes most likely are also effective in other regions and during other periods, as we chose a night with conditions typical of the monsoon season and two of the relevant processes - gravity waves and triggering of new clouds upstream of existing clouds - are rather independent of local orography. Nevertheless, the sensitivity of cloud formation to factors, such as orography, geographic location, variations of the LLJ and humidity distribution, model-grid spacing, model-domain size, turbulence parameterisation and aerosol-cloud interactions, which are believed to have a noticeable impact on cloud formation (Knippertz et al., 2015b), was not considered and could be investigated in the framework of the DACCIWA project.



*Acknowledgements.* The research leading to these results has received funding from the European Union 7th Framework Programme (FP7/2007-2013) under Grant Agreement no. 603502 (EU project DACCIWA: Dynamics-aerosol-chemistry-cloud interactions in West Africa). Special thanks go to Fabienne Lohou, Marie Lothon, Cheikh Dione, Peter Knippertz and Andreas Fink for their critical comments on the manuscript.



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





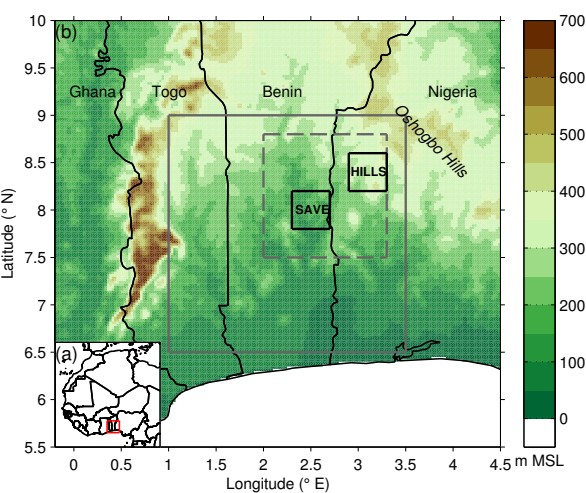

**Figure 1.** Location of the area of interest (red box) in West Africa (a) and orography of the area of interest in southern West Africa (b). The model domain and the area used for the analysis are indicated by the boxes with the solid grey and dashed grey lines, respectively, and the SAVE and HILLS boxes show the areas used for detailed investigation. The city of Savè is located in the centre of SAVE.





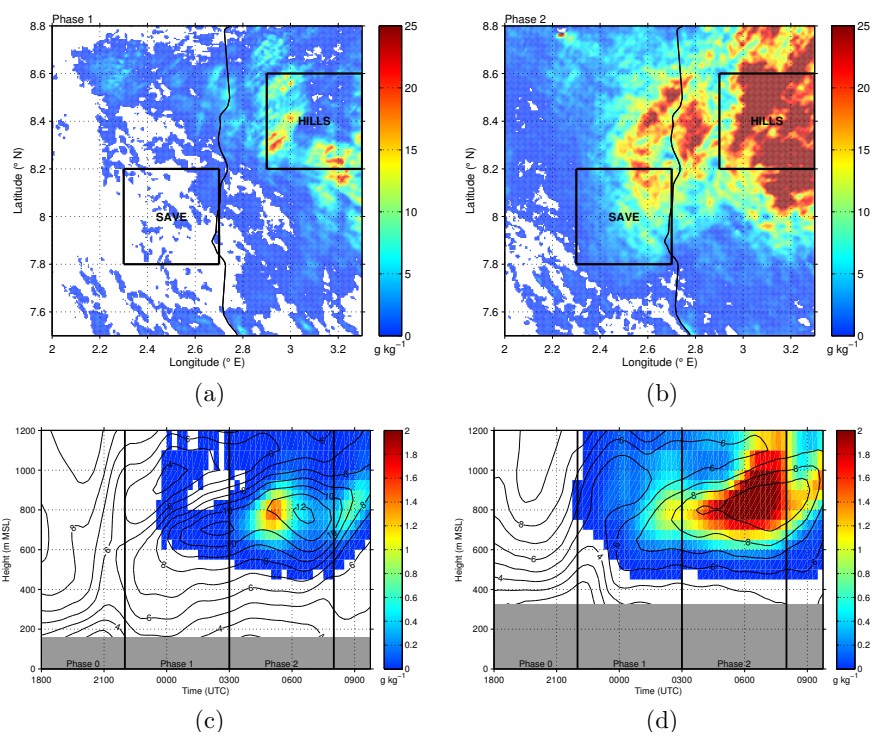

**Figure 2.** Spatial distribution of the liquid water content accumulated below 1000 m MSL averaged between 2200 and 0300 UTC (a; Phase 1) and between 0300 and 0800 UTC (b; Phase 2) and temporal evolution of the liquid water content (colour-coded) and horizontal wind in m s$^{-1}$ (black contours) profiles averaged for SAVE (c) and HILLS (d). Grey shaded areas in (c) and (d) mark the mean terrain height.





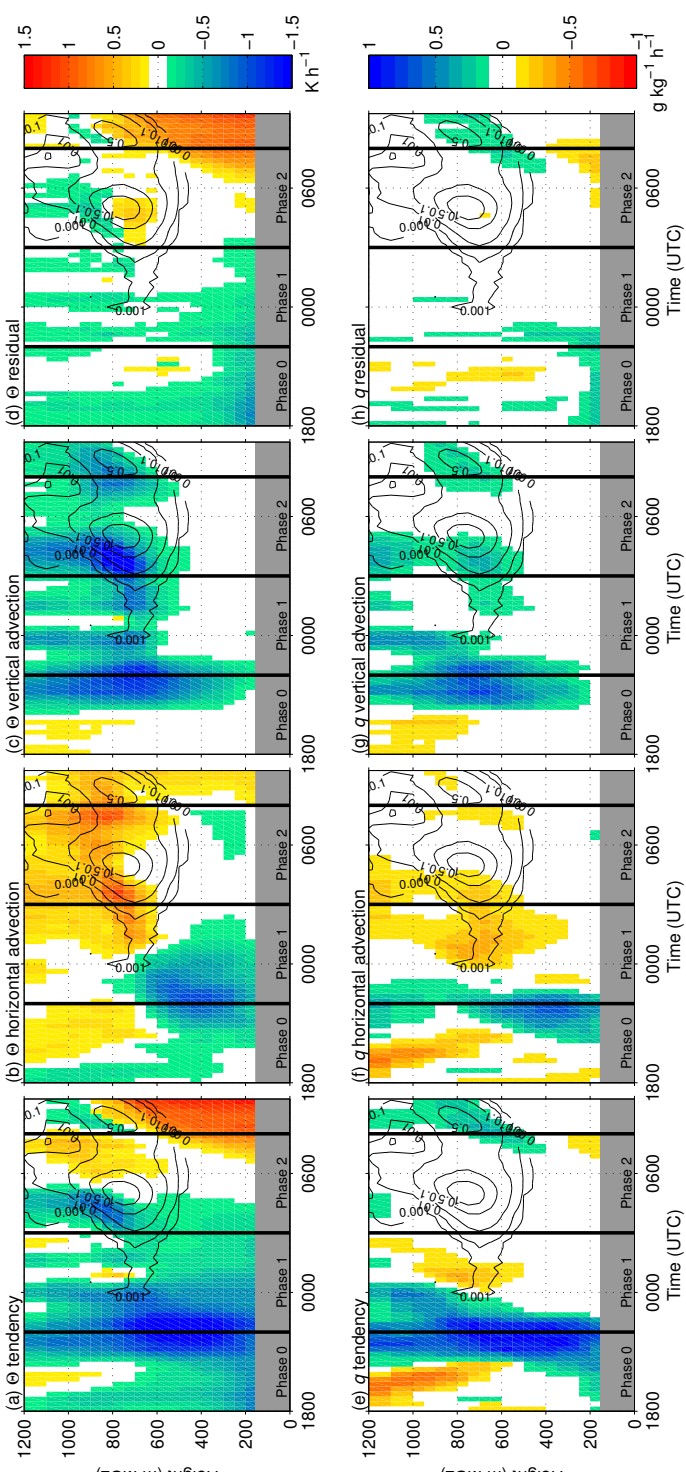

**Figure 3.** Temporal evolution of tendencies and of contributions to tendencies by horizontal advection and vertical advection for potential temperature (a-c) and moisture (specific humidity; e-g) averaged for SAVE. The residuals are determined by subtracting the contributions by advection from the tendencies (d, h). Liquid water content in g kg$^{-1}$ is indicated by the black contours. Grey shaded areas mark the mean terrain height.

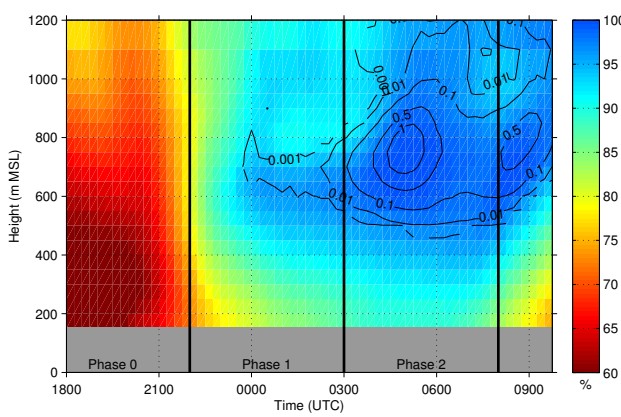

**Figure 4.** Temporal evolution of relative humidity (colour-coded) and liquid water content in g kg$^{-1}$ (black contours) averaged for SAVE. The grey shaded area marks the mean terrain height.





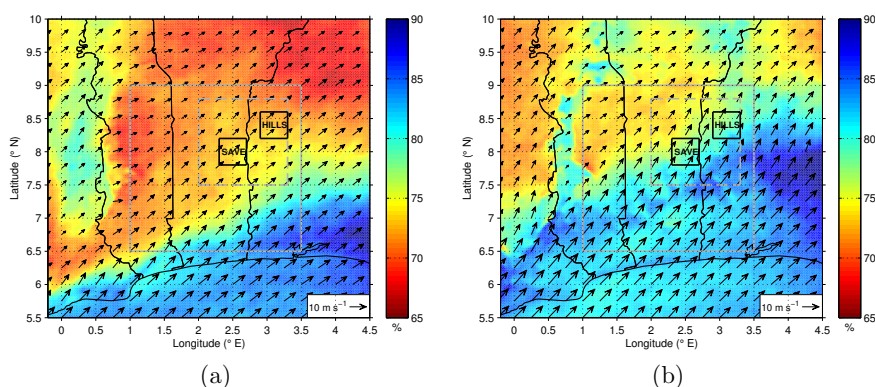

**Figure 5.** Spatial distribution of relative humidity (colour-coded) and horizontal wind (arrows) at 950 hPa at 1400 UTC (a) and 2100 UTC (b) averaged for the months of July and August 2006. Data are from the 2.8-km simulation.



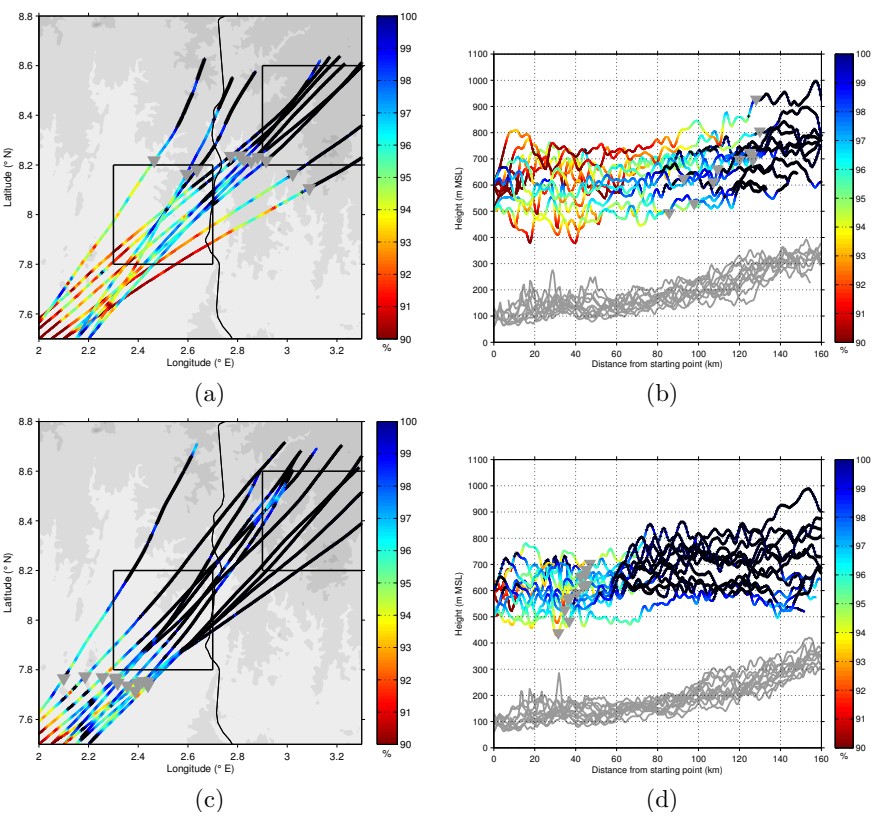

**Figure 6.** Relative humidity along trajectories of air parcels started at 7.5° N and between 1.9 and 2.2° E between 500 and 700 m MSL at 0000 UTC (a, b) and at 0200 UTC (c, d). Black markers indicate non-zero liquid water content and grey triangles indicate the beginning of Phase 2 for the respective air parcels. In (a) and (c), the boxes mark SAVE and HILLS, respectively, and terrain height is shown as grey shading with darker colour indicating higher terrain. In (b) and (d), grey lines indicate terrain height along the trajectories.





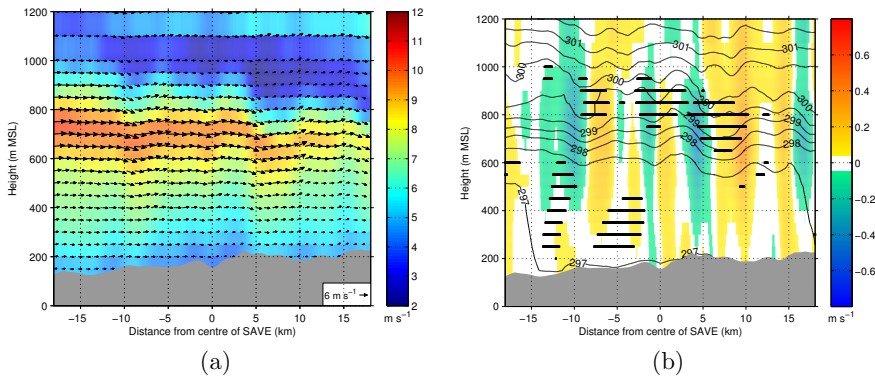

(a)            (b)

**Figure 7.** Vertical cross sections of wind speed (colour-coded) and flow components (horizontal and vertical (multiplied by 10); vectors) (a) and vertical wind speed (colour-coded) and potential temperature in K (black contours) (b) along a cross section through the centre of SAVE aligned along the mean wind direction of 207° at 0000 UTC. The mean wind direction was calculated for the layer below 1000 m MSL. In (b), black horizontal lines show areas with a turbulent kinetic energy higher than 0.5 m² s⁻². The grey shaded areas mark the terrain height.





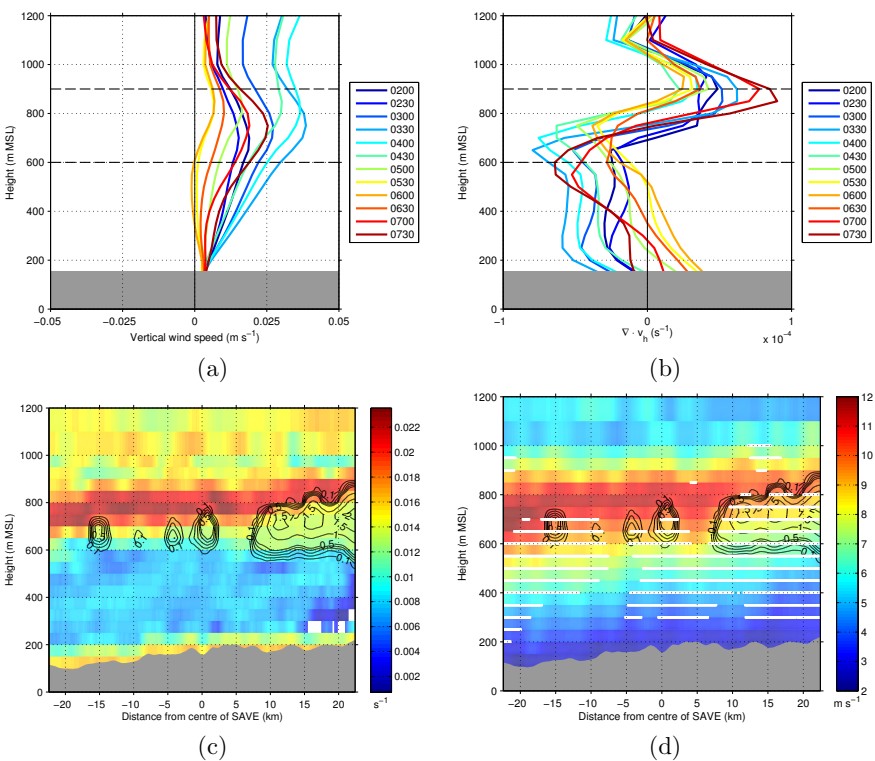

**Figure 8.** Profiles of vertical wind speed averaged for SAVE (a) and of horizontal wind divergence calculated for SAVE (b) at different times. Horizontal dashed lines enclose the layer where the liquid water content averaged for the shown period is larger than 0.01 g kg$^{-1}$. Vertical cross section of Brunt-Väisälä frequency (c) and horizontal wind speed (d) along a cross section through the centre of SAVE aligned along the mean wind direction of 223° at 0345 UTC. In (c) and (d), the mean wind direction is calculated for the layer below 1000 m MSL and black contours indicate liquid water content in g kg$^{-1}$. In (d), white horizontal lines show grid points with the turbulent kinetic energy higher than 0.5 m$^2$ s$^{-2}$. Grey shaded areas mark the terrain height.





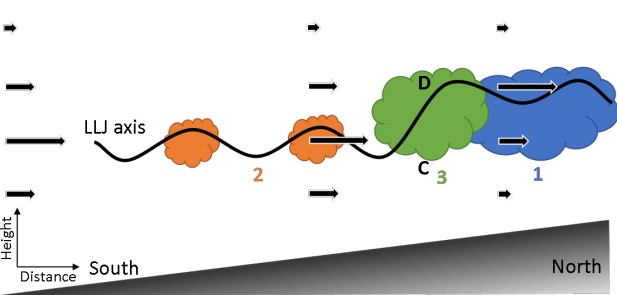

**Figure 9.** Schematic illustration of the processes contributing to cloud formation, i.e. orographically induced lifting (1), gravity waves (2) and horizontal convergence and upward motion upstream of existing clouds (3). Grey shading marks the terrain, arrows indicate the horizontal flow and the black line illustrates the height of the LLJ axis. C and D mark areas with horizontal convergence and divergence, respectively, related to process 3.