# Peer review of "Nocturnal low-level clouds over southern West Africa analysed using high-resolution simulations"

_Atmospheric Chemistry and Physics, 2016_

## Referee Comment (RC1) · Anonymous Referee #1 · 11 Nov 2016

This paper analyses the formation and evolution of low-level clouds which form in monsoon season in southern West Africa. From high resolution NWP simulations, the authors identify a front of high relative humidity which propagates inland overnight and drives the cloud formation, with uplift triggered by the orography and gravity waves. The paper is well written and interesting, and I believe suitable for publication subject to some minor amendments and further analysis.

Comments:

P3, L9-12 - these two sentences seem contradictory. You first state that a sub-grid cloud scheme is not applicable, and then go on to say that using one delays the transition of stratiform to convective clouds. Without any observations to compare against,

you cannot say whether this delay in convective initiation is a good or bad thing. Obviously a detailed discussion of the use of cloud schemes at ∼500m resolution is beyond the scope of this paper, but the authors should make it clear that the use of cloud schemes at this resolution is highly uncertain and it's not clear whether they are applicable or not (see, e.g. the discussion in Boutle et al. 2016). The authors can also use their observation of the difference in convective initiation as motivation for further study - DACCIWA presumably took the observations that will allow someone to analyse the timing of the stratiform to convective transition, and how important the cloud scheme is in this.

P3, L27 - is hourly really frequent enough to update the LBCs? I would have though ∼15 mins would be more appropriate. Have you tested the sensitivity of the simulations to this choice?

P4, L2 - why have you chosen to compute LWP below 1000m, when Figs 2 & 3 clearly show that the cloud extends above this height. Where are the clouds tops? Why not compute LWP up to the cloud top?

P6, L10 - a bit more discussion on why a phase lag of 90 degrees implies gravity waves, or a citation, would help the reader understand better what is going on here.

P9, L20-24 - given part of the motivation for the paper is informing the measurement strategy for DACCIWA, it would be good if you can give a bit more detail here on how these results have/did influence the measurements. What observations will be taken where, how will you track the relative humidity front, how will you observe the gravity waves, etc etc

Typos:

P1, L22 - I think it should just say 'MSL' in brackets

P2, L21 - should say "These results" instead of "This results"

P4, L4 - should say "allowing us to distinguish"

P9, L19 - should say "The simulations reveal"

References:

Boutle, I. A., Finnenkoetter, A., Lock, A. P. and Wells, H. (2016) The London Model: forecasting fog at 333m resolution. Q. J. R. Meteorol. Soc., 142, 360-371, doi:10.1002/qj.2656

---

## Referee Comment (RC2) · Anonymous Referee #2 · 30 Nov 2016

Review of "The life cycle of nocturnal low-level clouds over southern West Africa analysed using high-resolution simulations" by Bianca Adler, Norbert Kalthoff, and Leonard Gantner

Summary

This is a very well written study using high-resolution simulations with the COSMO model to investigate the processes that govern the evolution of low-level clouds in southern West Africa. The authors identify three processes, namely cooling of the atmosphere due to horizontal advection, gravity waves inducing vertical cold air advection, and enhanced convergence upstream of cloud formation triggering new clouds. The authors are careful to stress that these results consider a single case study and do

not all agree with previous studies for the same region, but use a thorough discussion of previous work in other regions to illustrate that these processes may well be relevant for low-level cloud formation in southern West Africa.

Brief verdict

I have no concerns with the scientific quality of this paper and believe the arguments and results are well presented. I have a list of comments that mostly regard confusing turns of phrases and a few points where the authors can add further clarification. I therefore recommend this paper be accepted with (very) minor corrections.

Specific comments

Title: To me, "life cycle" suggests the study of individual cloud elements from their origin to their decay, which is not what is covered in this paper. I would prefer "evolution", as it is more appropriate for a population of clouds. (Also title for section 3).

p.2 line 11: "a lot of effort is still needed" – At this point, the authors have not yet indicated why so much effort has gone into studying nocturnal low-level clouds over southern West Africa. One or two sentences regarding the motivation (weather? Climate?) should be introduced here.

p.2 line 20: "This results could then be used to optimize measurement strategy for the field campaign" – Since the field campaign has now taken place, the authors should rephrase this (assuming that their results have helped inform the strategy).

p.2 line 22-25: This is a nice way to distinguish this paper from previous studies.

p.2 line 32: "horizontal grid spacing of around 500 m" – has the COSMO model been verified and evaluated at this resolution? Perhaps not for southern West Africa, but one or two references covering COSMO at 500m would be useful here.

p.3 line 10: "sensitivity tests show" – Do you have a reference for this?

p.3 line 13-14: "online trajectory module" – At this point, could you add an additional

sentence stating the purpose of this module for your work. What are you hoping to learn from these trajectories?

p.3 line 15-21: This is a useful paragraph building the reader's confidence in the simulation.

p.3 line 26: "we choose the night from 3 to 4 August for analysis" – This is slightly confusing, as in line 19 it says "the simulated conditions on a specific day do not necessarily agree with the observations". So are these dates relevant? Perhaps the authors could clarify for the reader again that these dates refer to the model simulation (unless the authors can confirm that also based on observations, these are appropriate days for their study).

p.4 line 3: "we accumulate the liquid water content" – so this is liquid water path?

p.4 line 3-4: Is 15-minute output sufficient to study the temporal evolution of low-level clouds, when individual cloud elements likely don't last more than 30 minutes? This is more a comment on using "life cycle".

p.4 line 6: "cloud-free" – Can the authors confirm there is no ice cloud aloft?

p.4 line 11: "clouds... are more intense" – what does this mean? That the clouds have higher LWC?

p.4 line 12: "very similar" – in this sentence structure, I would suggest "remarkably similar".

p.4 line 17, 19: "the processes", "the different budget terms" – Could you please be more specific. Which budgets matter for your study? Which processes can be expected to affect the atmospheric conditions?

Title of section 4.1 "Before the onset..."

p.5 line 6: "meridional averaging" – is this not domain averaging, i.e. over the SAVE box?

p.5 line 11-23: Does the front differ much in structure in the 500m simulation for the single case study?

p.5 line 30: "once the front has passed SAVE" – please specify at what time, especially referring to figure 3.

p.5 line 32: "weak horizontal dry air advection" – how significant (important) is this? It seems rather patchy in figure 3.

p.6 line 4-5: "gravity waves... Fig(6)" – Please clarify in the main text how the gravity waves can be identified in figure 6. Presumably it's the wavelike pattern of the trajectories?

p.6 line 5: "broken band-like structures" – this is not obvious from the figures referenced.

Title of section 4.3 "Intensification" – in what sense? Thicker clouds? Higher LWC? I don't know how to interpret an "intense cloud".

p.6 line 17: "thick" – geometrically? Optically? Occurs also on p.7 lines 5 and 6.

p.6 line 19: "respectively" – this could refer to "phases 0 and 1" in this sentence structure. Perhaps best to disentangle the sentence to say "in the clouds is continuously cooled by vertical advection and below the clouds by horizontal advection" (or something similar/better).

p.6 line 22-23: "mean upward motion" – (perhaps obvious) what causes the mean upward motion?

p. 6 line 26: "intensity" – see earlier comment.

p.7 line 6 and line 11: "spread", "retreat" – how does this happen? Do new clouds form? Do clouds propagate or advect with the flow? Do clouds grow?

p.7 line 27-28: "this simulation" – which simulation? The COSMO 500m simulation of

the current paper, or the Schuster et al. (2013) simulation?

p.7 line 32: "this process has not been reported before" – Anywhere? Or just for this particular region in southern West Africa?

p.8 line 7: Specify that Schuster et al. (2013) used simulations and Van der Linden et al. (2015) used observations.

p.8 line 9: "latter" – I'm not sure which process is referred to here, perhaps rephrase the sentence so that "latter" does not need to be used.

p.8 line 32-33: "In the opinion of the authors" – Who? Grams et al.? Please rephrase.

———————————————————————

---

## Author Comment (AC1) · 2 Jan 2017

**Response to the Referees (ACP-2016-842)**

**"The life cycle of nocturnal low-level clouds over southern West Africa analysed using high-resolution simulations" by Bianca Adler, Norbert Kalthoff and Leonhard Gantner**

January 2, 2017

We thank the two anonymous reviewers for their clear and helpful comments. We respond to all comments of the reviewer in this document and we prepared a revised manuscripts where major changes are marked in red. In the following, comments of the reviewers are given in italic style, while our responses are given in normal font style.

**1 Anonymous Referee #1**

*This paper analyses the formation and evolution of low-level clouds which form in monsoon season in southern West Africa. From high resolution NWP simulations, the authors identify a front of high relative humidity which propagates inland overnight and drives the cloud formation, with uplift triggered by the orography and gravity waves. The paper is well written and interesting, and I believe suitable for publication subject to some minor amendments and further analysis.*

Comments:

1. *P3, L9-12 - these two sentences seem contradictory. You first state that a sub-grid cloud scheme is not applicable, and then go on to say that using one delays the transition of stratiform to convective clouds. Without any observations to compare against you cannot say whether this delay in convective initiation is a good or bad thing. Obviously a detailed discussion of the use of cloud schemes at 500 m resolution is beyond the scope of this paper, but the authors should make it clear that the use of cloud schemes at this resolution is highly uncertain and its not clear whether they are applicable or not (see, e.g. the discussion in Boutle et al. 2016). The authors can also use their observation of the difference in convective initiation as motivation for further study - DACCIWA presumably took the observations that will allow someone to analyse the timing of the stratiform to convective transition, and how important the cloud scheme is in this.*

**Response:** The phrasing in the manuscript was misleading. Indeed, we do not know if the sub-grid scale cloud scheme applies for the used resolution of 500 m. We can only speculate that the criteria developed for simulations with coarser grid spacing do not apply for simulations with higher horizontal resolutions. Boutle et al. 2016 investigated the potential of a very high resolution model for forecasting fog over the south-eastern UK and found a very strong dependency of the results on the critical relative humidity used for the cloud parametrisation. They also state that the necessity of using sub-grid scale cloud schemes crucially depends on the regime. As we cannot be sure if we should use the sub-grid scale cloud schemes for the conditions in West Africa we performed several simulations using a relative humidity criterion and a statistical criterion for sub-grid scale clouds and also using no sub-grid scale cloud scheme. We found a strong impact on the transition of stratiform to convective clouds while the impact on the nocturnal stratus was small. We decided to use no sub-grid scale cloud scheme, keeping in mind that we do not know if this leads to more realistic results or not. We agree with the reviewer that the observations during DACCIWA could be valuable to investigate the impact of a sub-grid scale cloud scheme on stratiform and convective clouds in high-resolution models and we added this to the manuscript. To clarify we rewrote this part of the manuscript:

"However, the consideration of sub-grid scale clouds is highly uncertain at such high horizontal resolution as chosen in this study and it is not clear whether the criteria are applicable or not (Boutle et al., 2016). We performed sensitivity tests for the region in West Africa and found that the consideration of sub-grid-scale clouds in the radiation scheme delays the transition from stratiform clouds to convective clouds by several hours, while the impact on the qualitative characteristics of the nocturnal low-level clouds is small. For the simulation analysed in this study, we decided to consider only grid-scale clouds in the radiation scheme, which is called every 15 min. Whether this choice leads to more realistic results or not, could be investigated in upcoming studies using observations from the DACCIWA field campaign."

2. *P3, L27 - is hourly really frequent enough to update the LBCs? I would have thought 15 mins would be more appropriate. Have you tested the sensitivity of the simulations to this choice?*

   **Response:** As the output of the 2.8 km simulation which we used for initialisation and boundary conditions was hourly, we could not investigate the sensitivity of the 500 m simulations to the frequency of the boundary conditions updates. However, the large-scale conditions were rather homogeneous during the simulated period (no MCS or other disturbances) and we do not expect major changes in horizontal advection at the boundaries throughout the night, as the front did not enter through the boundaries but existed within the model domain.

3. *P4, L2 - why have you chosen to compute LWP below 1000m, when Figs 2 & 3 clearly show that the cloud extends above this height. Where are the clouds tops? Why not compute LWP up to the cloud top?*

   **Response:** Our decision to average the liquid water content below 1000 m, was based on averaged profiles of liquid water content for SAVE. In this area there is clear minimum in liquid water content at 1000 m MST, which indicates the top of the low-level clouds. As the

[Figure]

Figure 1: Hourly profiles of liquid water content averaged between 2° and 3.3° E and 7.5° and 8.8° N.

values of averaged liquid water content are not zero, this minimum is hard to distinguish in Fig. 2c due to the colour scale. Because of the reviewer's comment we looked again at the data and plotted profiles averaged for the whole area of interest (area shown in Figs. 2a, b) and found that the minimum of mean liquid water content for this area was rather at 1200 m than at 1000 m (Fig. 1). Thus, we replaced Figs. 2a, b with the liquid water content accumulated below 1200 m MSL and added the information to the text why we chose this layer for the liquid water content accumulation.

"To obtain information about the spatial distribution and temporal evolution of low-level clouds, we accumulate the liquid water content up to 1200 m MSL, i.e. up to the top of the domain-averaged low-level clouds, (Figs. 2a, b) and average profiles of the liquid water content for both areas SAVE (Fig. 2c) and HILLS (Fig. 2d)."

4. *P6, L10 - a bit more discussion on why a phase lag of 90 degrees implies gravity waves, or a citation, would help the reader understand better what is going on here.*

**Response:** The phase shift in the fluctuations of potential temperature and vertical wind speed is 90° for gravity waves. This is for example stated in Durran (1990). For convection these fluctuations are in phase. This phase shift in gravity waves can be explained as follows: when the fluctuations of potential temperature are largest, i.e. at the air parcels maximum positive or negative displacement, the vertical wind speed is zero. Between these locations the air parcels are either gaining or loosing buoyancy, reaching their maximum vertical wind speed when variations of potential temperature are smallest. We added the reference to Durran (1990) to the manuscript.

"Fast Fourier Transformation reveals a phase lag of roughly 90° between the waves of potential temperature and vertical wind speed, which is a characteristic of gravity waves (Durran,

1990),..."

5. *P9, L20-24 - given part of the motivation for the paper is informing the measurement strategy for DACCIWA, it would be good if you can give a bit more detail here on how these results have/did influence the measurements. What observations will be taken where, how will you track the relative humidity front, how will you observe the gravity waves, etc etc*

   **Response:** We added details about specific measurements taken during the field campaign to the text:

   "For example, wind profiles were measured with high temporal and vertical resolution using radiosondes, sodars and a UHF profiler. Research aircrafts were flying inland from the coast detecting horizontal gradients in e.g. temperature and moisture. A combination of vertical stare and scanning mode was chosen for a doppler lidar system to allow for detection of inhomogeneities in wind and backscatter in an area of up to 20 km x 20 km, which could provide evidence for the front, as well as for vertical velocity, which can be used as an indicator for gravity waves and to derive turbulence profiles."

**Typos:**

1. *P1, L22 - I think it should just say MSL in brackets*

   We changed the text accordingly.

2. *P2, L21 - should say "These results" instead of "This results"*

   We changed the text accordingly.

3. *P4, L4 - should say "allowing us to distinguish"*

   We changed the text accordingly.

4. *P9, L19 - should say "The simulations reveal"*

   We changed the text to "The simulation reveals" as we mainly analysed one simulation.

**2 Anonymous Referee #2**

*This is a very well written study using high-resolution simulations with the COSMO model to investigate the processes that govern the evolution of low-level clouds in southern West Africa. The authors identify three processes, namely cooling of the atmosphere due to horizontal advection, gravity waves inducing vertical cold air advection, and enhanced convergence upstream of cloud formation triggering new clouds. The authors are careful to stress that these results consider a*

*single case study and do not all agree with previous studies for the same region, but use a thorough discussion of previous work in other regions to illustrate that these processes may well be relevant for low-level cloud formation in southern West Africa.*

*I have no concerns with the scientific quality of this paper and believe the arguments and results are well presented. I have a list of comments that mostly regard confusing turns of phrases and a few points where the authors can add further clarification. I therefore recommend this paper be accepted with (very) minor corrections.*

**Specific comments**

1. *Title: To me, "life cycle" suggests the study of individual cloud elements from their origin to their decay, which is not what is covered in this paper. I would prefer "evolution", as it is more appropriate for a population of clouds. (Also title for section 3).*

   **Response:** We agree with the reviewer that life-cycle suggests the study of individual clouds. To solve this problem, we decided to remove life-cycle from the title, modified the title of Section 3 and changed the text accordingly by replacing "life cycle" with "evolution/development".

   "Nocturnal low-level clouds over southern West Africa analysed using high-resolution simulations"

   "Evolution of low-level clouds"

2. *p.2 line 11: "a lot of effort is still needed" – At this point, the authors have not yet indicated why so much effort has gone into studying nocturnal low-level clouds over southern West Africa. One or two sentences regarding the motivation (weather? Climate?) should be introduced here.*

   **Response:** We added a sentence at the beginning of the introduction to explain how the clouds affect the atmospheric conditions and why they are important:

   "As these clouds persist into the late morning or early afternoon, they reduce surface solar radiation and could strongly affect the regional heat and moisture budgets and, thus, the West African climate (Knippertz et al., 2011)."

3. *p.2 line 20: "This results could then be used to optimise the measurement strategy for the field campaign" – Since the field campaign has now taken place, the authors should rephrase this (assuming that their results have helped inform the strategy).*

   **Response:** We rephrased the sentence:

   "These results were then used to optimise the measurement strategy for the field campaign."

4. *p.2 line 22-25: This is a nice way to distinguish this paper from previous studies.*

   **Response:** Thank you.

5. *p.2 line 32: "horizontal grid spacing of around 500 m" – has the COSMO model been verified and evaluated at this resolution? Perhaps not for southern West Africa, but one or two references covering COSMO at 500m would be useful here.*

   **Response:** We added references to the text:

   "COSMO model simulations with similar high horizontal resolution have been performed by Fiori et al. (2010) for studying deep moist convection and by Gantner et al. (2016) to investigate the cloud-topped boundary layer."

6. *p.3 line 10: "sensitivity tests show" – Do you have a reference for this?*

   **Response:** The phrasing in the manuscript was misleading. As we do not know if the sub-grid scale cloud scheme applies for the used high horizontal resolution, we performed several simulations using a relative humidity criterion and a statistical criterion for sub-grid scale clouds and also using no sub-grid scale cloud scheme. We rephrased this part of the manuscript (also according to Comment 1 of Reviewer #1) and it now reads as follows:

   "However, the consideration of sub-grid scale clouds is highly uncertain at such high horizontal resolution as chosen in this study and it is not clear whether the criteria are applicable or not (Boutle et al., 2016). We performed sensitivity tests for the region in West Africa and found that the consideration of sub-grid-scale clouds in the radiation scheme delays the transition from stratiform clouds to convective clouds by several hours, while the impact on the qualitative characteristics of the nocturnal low-level clouds is small. For the simulation analysed in this study, we decided to consider only grid-scale clouds in the radiation scheme, which is called every 15 min. Whether this choice leads to more realistic results or not, could be investigated in future studies using observations from the DACCIWA field campaign."

7. *p.3 line 13-14: "online trajectory module" – At this point, could you add an additional sentence stating the purpose of this module for your work. What are you hoping to learn from these trajectories?*

   **Response:** We added the information about the purpose of this module to the text:

   "Using an online trajectory module implemented in COSMO (Miltenberger et al., 2013), trajectories are started hourly at 7.5° N between 1.5° and 2.5° E at various levels below 2000 m MSL in order to obtain information on the origin of air parcels which are involved in the evolution of the low-level clouds."

8. *p.3 line 15-21: This is a useful paragraph building the readers confidence in the simulation.*

   **Response:** Thank you.

9. *p.3 line 26: "we choose the night from 3 to 4 August for analysis" This is slightly confusing, as in line 19 it says "the simulated conditions on a specific day do not necessarily agree with the observations". So are these dates relevant? Perhaps the authors could clarify for the reader again that these dates refer to the model simulation (unless the authors can confirm that also based on observations, these are appropriate days for their study).*

   **Response:** As the exact dates are not relevant we removed them from the manuscript in order to avoid confusion. Instead we added the information that the simulation was done for a night at the beginning of August and run for 30 hours.

   " Of these periods, we choose a night at the beginning of August for analysis. The simulation is initialised at 1200 UTC and runs for 30 h with the boundary conditions being updated every hour. "

10. *p.4 line 3: "we accumulate the liquid water content" so this is liquid water path?*

    **Response:** Liquid water path is a measure of the total amount of liquid water in a vertical column of the atmosphere extending from the surface to the top of the atmosphere. Because we only accumulate up to a certain height and because liquid clouds are present above this certain height, the quantity shown in Fig. 2a, b is not the liquid water path.

11. *p.4 line 3-4: Is 15-minute output sufficient to study the temporal evolution of low-level clouds, when individual cloud elements likely dont last more than 30 minutes? This is more a comment on using "life cycle".*

    **Response:** We agree that 15-min output would not be sufficient to study the life-cycle of individual clouds. For our purpose, however, namely to study the evolution of the low-level clouds the 15-min output is sufficient. Please also see response to comment 1.

12. *p.4 line 6: "cloud-free" – Can the authors confirm there is no ice cloud aloft?*

    **Response:** We checked profiles of liquid water content and ice content up to higher levels and found liquid water clouds at around 5 km and ice clouds at around 10 km height at 220 UTC. We rephrased the sentence in the manuscript:

    "Before about 2200 UTC, the analysed area is free of low-level clouds (Phase 0)."

13. *p.4 line 11: "clouds. . . are more intense" – what does this mean? That the clouds have higher LWC?*

    **Response:** With intense clouds, we intend to say that the clouds have a higher liquid water content and a larger vertical extent. We understand that this expression is not clear and modified it in the text.

    "Throughout the night, the horizontal and vertical extent as well as the liquid water content of the low-level clouds vary, allowing us to distinguish different phases."

"During this phase, the clouds in HILLS have a higher density and a larger vertical extent than during Phase 1 "

14. *p.4 line 12: "very similar" – in this sentence structure, I would suggest remarkably similar.*

   **Response:** We changed the text accordingly.

15. *p.4 line 17, 19: "the processes", "the different budget terms" – Could you please be more specific. Which budgets matter for your study? Which processes can be expected to affect the atmospheric conditions?*

   **Response:** We added more details to the text:

   "To understand the evolution of low-level clouds, we analyse in detail the atmospheric conditions and the processes such as advection, which affect these conditions during the different phases."

   "To study contributions to the tendencies by advection, we determine horizontal and vertical advection for each grid point and each level and average over the area."

16. *Title of section 4.1 "Before the onset. . ."*

   **Response:** We changed the text accordingly.

17. *p.5 line 6: "meridional averaging" – is this not domain averaging, i.e. over the SAVE box?*

   **Response:** We averaged over the box, but because the front is west-east oriented the meridional averaging is mostly responsible for the smoothing. To avoid confusion we rephrased the sentence.

   "Due to box averaging, cooling and moistening appears to be rather smooth."

18. *p.5 line 11-23: Does the front differ much in structure in the 500m simulation for the single case study?*

   **Response:** The structure of the front in the 500 m simulation is very similar with respect to timing, orientation and absolute values to the averaged one. On p.5 l. 22 we state that the front reaches Save at around 2100 UTC on the average, which agrees well with the conditions of the case study .

19. *p.5 line 30: "once the front has passed SAVE" - please specify at what time, especially referring to figure 3.*

   **Response:** We added the information that front has passed SAVE at around 2300 UTC to the text.

20. *p.5 line 32: "weak horizontal dry air advection" – how significant (important) is this? It seems rather patchy in figure 3.*

**Response:** The horizontal dry air advection below the cloud layer is indeed quite weak but existent. Values are between -0.05 g kg$^{-1}$ h$^{-1}$ and -0.15 g kg$^{-1}$ h$^{-1}$. In Fig. 3 values above -0.1 g kg$^{-1}$ h$^{-1}$ are plotted white for better clarity. This is why the dry air advection below the cloud layer appears patchy in the figure. Even if the dry air advection is weak in this case we cannot exclude that it might play an important role in other cases and might even inhibit the evolution of low-level clouds.

21. *p.6 line 4-5: "gravity waves. . . Fig(6)" – Please clarify in the main text how the gravity waves can be identified in figure 6. Presumably its the wavelike pattern of the trajectories?*

    **Response:** We added the information to the text that the gravity waves are indicated by the wavelike structure of the trajectories.

    "Besides orographically induced lifting, which is less strong in SAVE than in HILLS due to different terrain gradients, gravity waves indicated by the wavelike structure of the trajectories contribute to the upward motion in the nocturnal atmosphere (Fig. 6)."

22. *p.6 line 5: "broken band-like structures" – this is not obvious from the figures referenced.*

    **Response:** In Fig. 6c black markers indicate bands of cloud aligned perpendicular to the trajectories. While we think that this is obvious in this figure, we agree with the reviewer that the bands of clouds in Fig. 2a are a little harder to see, as they appear broken and patchy. We added the information that the black markers indicate the clouds in Fig. 6c.

    "During Phase 1, saturation is reached in wave crests and clouds form in broken band-like structures perpendicular to the mean flow in some regions in the south-western part of the analysed area (Fig. 2a and black markers in Fig. 6c)."

23. *Title of section 4.3 "Intensification" – in what sense? Thicker clouds? Higher LWC? I dont know how to interpret an "intense cloud".*

    **Response:** This comment relates to comment 13. As stated there we replaced intense with higher liquid water content and larger vertical extent. We now use the terms high density and low density to describe clouds with high and low liquid water content. To clarify this we added a sentence to the beginning of Section 3. We also replaced intensification in the title of 4.3 with "Increase of density".

    "In the following, we use the term density to distinguish between clouds with low and high liquid water content."

    "Increase of density and spatial expansion of clouds"

24. *p.6 line 17: "thick" – geometrically? Optically? Occurs also on p.7 lines 5 and 6.*

    **Response:** See responses to comments 13 and 23. We now use dense for clouds with a high liquid water content.

25. *p.6 line 19: "respectively" – this could refer to "phases 0 and 1" in this sentence structure. Perhaps best to disentangle the sentence to say "in the clouds is continuously cooled by vertical advection and below the clouds by horizontal advection" (or something similar/better).*

**Response:** We rephrased the sentence:

"We assume that two processes are mainly responsible for this: (i) during Phases 0 and 1, the atmosphere below the clouds is continuously cooled by horizontal advection while vertical cold air advection mainly affects the layer where clouds form (Figs. 3a, b, c)."

26. *p.6 line 22-23: "mean upward motion" – (perhaps obvious) what causes the mean upward motion?*

**Response:** The reason for the mean upward motion is the shift of the wind speed maximum to the cloud top, which causes horizontal convergence upstream of the clouds. This process is explained in detail in the paragraph from p.6 l.26 to p.7 l.5.

27. *p. 6 line 26: "intensity" – see earlier comment.*

**Response:** Here intensity refers to the gravity waves. To clarify this we replaced intensity with amplitude and frequency:

"The sudden stronger upward motion cannot be caused by gravity waves, because their amplitude and frequency remain about the same."

28. *p.7 line 6 and line 11: "spread2, "retreat" – how does this happen? Do new clouds form? Do clouds propagate or advect with the flow? Do clouds grow?*

**Response:** The clouds spread to the upstream side, i.e. new clouds form. The process responsible for this is the shift of the wind speed maximum to the cloud top, which causes horizontal convergence upstream of the clouds. This is explained in detail in the text (p.6 l.26 to p.7 l.5). As stated at p.7 l.7-8 relative humidity is several % lower southwest of SAVE. When the upstream border of low-level cloud deck reaches this region the process triggering new clouds upstream of existing ones is not able to produce new clouds. The cloud deck slowly dissolves from the southwest. To clarify what we mean with retreat we added a sentence to the text:

"Once the south- western border of the low-level cloud deck reaches the area with lower relative humidity at around 0500 UTC, no new clouds are able to form and the clouds slowly dissolve from the south-west and gradually retreat to the north-east."

29. *p.7 line 27-28: "this simulation" which simulation? The COSMO 500m simulation of the current paper, or the Schuster et al. (2013) simulation?*

**Response:** To clarify this we changed "this simulation" to "our simulation/this present simulation":

"The strength and height of the LLJ are comparable to the LLJ characteristics during cloudy nights reported by Schrage and Fink (2012) and Schuster et al. (2013) considering that the LLJ and cloud base in our simulation are roughly constant in height above sea level. The low-level clouds in the present simulation first form directly upstream of the Oshogbo Hills due to orographically induced lifting and then spread to the south-west, i.e. to the upstream side."

30. *p.7 line 32: "this process has not been reported before" – Anywhere? Or just for this particular region in southern West Africa?*

    **Response:** As far as we know, this process has not been reported anywhere before:

    "According to our knowledge, this process has not been reported in any region before."

31. *p.8 line 7: Specify that Schuster et al. (2013) used simulations and Van der Linden et al. (2015) used observations.*

    **Response:** We added this to the text:

    "Besides the formation of low-level clouds over land upstream of orography, it is found in simulations (Schuster et al., 2013) and satellite observations (van der Linden et al., 2015) that low-level clouds form early along the Guinea Coast and then spread inland."

32. *p.8 line 9: "latter" – I'm not sure which process is referred to here, perhaps rephrase the sentence so that "latter" does not need to be used.*

    **Response:** We rephrased the sentence:

    " The model-domain size and location in this study do not allow to study cloud formation along the coast in detail."

33. *p.8 line 32-33: "In the opinion of the authors" – Who? Grams et al.? Please rephrase.*

    **Response:** We rephrased the sentence:

[revised manuscript text omitted]